# TRANSFORMERS FOR MODELING PHYSICAL SYSTEMS

## ABSTRACT

Transformers are widely used in neural language processing due to their ability to model longer-term dependencies in text. Although these models achieve state-of-the-art performance for many language related tasks, their applicability outside of the neural language processing field has been minimal. In this work, we propose the use of transformer models for the prediction of dynamical systems representative of physical phenomena. The use of Koopman based embeddings provide a unique and powerful method for projecting any dynamical system into a vector representation which can then be predicted by a transformer model. The proposed model is able to accurately predict various dynamical systems and outperform classical methods that are commonly used in the scientific machine learning literature.[1] [2]

## 1 INTRODUCTION

The transformer model (Vaswani et al., 2017), built on self-attention, has largely become the state-of-the-art approach for a large set of neural language processing (NLP) tasks including language modeling, text classification, question answering, etc. Although more recent transformer work is focused on unsupervised pre-training of extremely large models (Devlin et al., 2018; Radford et al., 2019; Dai et al., 2019; Liu et al., 2019), the original transformer model garnered attention due to its ability to out-perform other state-of-the-art methods by learning longer-term dependencies without recurrent connections. Given that the transformer model was originally developed for NLP, nearly all related work has been rightfully confined within this field with only a few exceptions. Here, we focus on the development of transformers to model dynamical systems that can replace otherwise expensive numerical solvers. In other words, we are interested in using transformers to learn the language of physics.

The surrogate modeling of physical systems is a research field that has existed for several decades and is a large ongoing effort in scientific machine learning. A surrogate model is defined as an approximate model of a physical phenomenon that is designed to replace an expensive computational solver that would otherwise be needed to resolve the system of interest. The key characteristic of surrogate models is their ability to model a distribution of initial or boundary conditions rather than learning just one solution. This is arguably essential for the justification of training a deep learning model versus using a standard numerical solver. The most tangible applications of surrogates are for optimization, design and inverse problems where many repeated simulations are typically needed. With the growing interest in deep learning, deep neural networks have been used for surrogate modeling a large range of physical systems in recent literature.

Standard deep neural network architectures such as auto-regressive (Mo et al., 2019; Geneva & Zabaras, 2020a), residual/Euler (González-García et al., 1998; Sanchez-Gonzalez et al., 2020), recurrent and LSTM based models (Mo et al., 2019; Tang et al., 2020; Maulik et al., 2020) have been largely demonstrated to be effective at modeling various physical dynamics. Such models generally rely exclusively on the past time-step to provide complete information on the current state of the system's evolution. Particularly for dynamical systems, present machine learning models lack generalizable time cognisant capabilities to predict multi-time-scale phenomena present in systems including turbulent fluid flow, multi-scale materials modeling, molecular dynamics, chemical processes, etc. Thus currently adopted models struggle to maintain true physical accuracy for long-time

---

[1]Code available at: [URL available after review].
[2]Supplementary videos available at: https://sites.google.com/view/transformersphysx.

predictions. Much work is needed to scale such deep learning models to larger physical systems that are of scientific and industrial interest. This work deviates from this pre-existing literature by investigating the use of transformers for the prediction of physical systems, relying entirely on self-attention to model dynamics. In the recent work of Shalova & Oseledets (2020), such self-attention models were tested to learn single solutions of several low-dimensional ordinary differential equations. In this work, we propose a physics inspired embedding methodology to model a distribution of dynamical solutions. We will demonstrate our model on high-dimensional partial differential equation problems that far surpass the complexity of past works. To the authors best knowledge, this is the first work to explore transformer NLP architectures for the prediction of physical systems.

## 2 METHODS

When discussing dynamical systems, we are interested in systems that can be described through a dynamical ordinary or partial differential equation:

$$\phi_t = F\left(\boldsymbol{x}, \phi(t, \boldsymbol{x}, \boldsymbol{\eta}), \nabla_{\boldsymbol{x}}\phi, \nabla_{\boldsymbol{x}}^2\phi, \phi \cdot \nabla_{\boldsymbol{x}}\phi, \dots\right), \quad F : \mathbb{R} \times \mathbb{R}^n \to \mathbb{R}^n,$$
$$t \in \mathcal{T} \subset \mathbb{R}^+, \quad \boldsymbol{x} \in \Omega \subset \mathbb{R}^m,$$

(1)

in which $\phi \in \mathbb{R}^n$ is the solution of this differential equation with parameters $\boldsymbol{\eta}$, in the time interval $\mathcal{T}$ and spatial domain $\Omega$ with a boundary $\Gamma \subset \Omega$. This general form can embody a vast spectrum of physical phenomena including fluid flow and transport processes, mechanics and materials physics, and molecular dynamics. In this work, we are interested in learning the set of solutions for a distribution of initial conditions $\phi_0 \sim p(\phi_0)$, boundary conditions $\mathcal{B}(\phi) \sim p(\mathcal{B}) \,\forall \boldsymbol{x} \in \Gamma$ or equation parameters $\boldsymbol{\eta} \sim p(\boldsymbol{\eta})$. This accounts for modeling initial value, boundary value and stochastic problems. We emphasize that this is fundamentally different, more difficult and of greater interest for most scientific applications compared to learning a single solution.

To make this problem applicable to the use of transformer models, the continuous solution is discretized in both the spatial and temporal domains such that the solution of the differential equation is $\Phi = \{\phi_0, \phi_1, \dots \phi_T\}; \phi_i \in \mathbb{R}^{n \times d}$ for which $\phi_i$ has been discretized by $d$ points in $\Omega$. We assume an initial state $\phi_0$ and that the time interval $\mathcal{T}$ is discretized by $T$ time-steps with a time-step size $\Delta t$. Hence, we pose the problem of modeling a dynamical system as a time-series problem. The machine learning methodology has two core components: the transformer for modeling dynamics and the embedding network for projecting physical states into a vector representation. Similar to NLP, the embedding model is trained prior to the transformer. This embedding model is then frozen and the entire data-set is converted to the embedded space in which the transformer is then trained as illustrated in Fig. 1. During testing, the embedding decoder is used to reconstruct the physical states from the transformer's predictions.

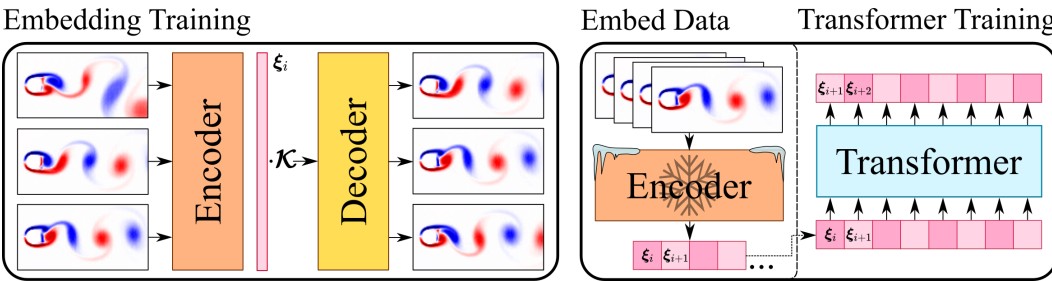

Figure 1: The two training stages for modeling physical dynamics using transformers. (Left to right) The embedding model is first trained using Koopman based dynamics. The embedding model is then frozen (fixed), all training data is embedded and the transformer is trained in the embedded space.

### 2.1 TRANSFORMER

The transformer model was originally designed with NLP as the sole application with word vector embeddings of a passage of text being the primary input (Vaswani et al., 2017). However, recent works have explored using attention mechanisms for different machine learning tasks (Veličković et al., 2017; Zhang et al., 2019; Fu et al., 2019) and a few investigate the use of transformers for applications outside of the NLP field (Chen et al., 2020). This suggests that self-attention and in

particular transformer models may work well for any problem that can be posed as a sequence of vectors. In this work, the primary input to the transformer will be an embedded dynamical system, $\Xi = \{\boldsymbol{\xi}_0, \boldsymbol{\xi}_1, \ldots \boldsymbol{\xi}_T\}$, where the embedded state at time-step $i$ is denoted as $\boldsymbol{\xi}_i \in \mathbb{R}^e$. Given that we are interested in the prediction of a physical time series, this motivates the usage of a language modeling architecture that is designed for the sequential prediction of words in a body of text. We select the transformer decoder architecture used in the GPT models (Radford et al., 2018; 2019).

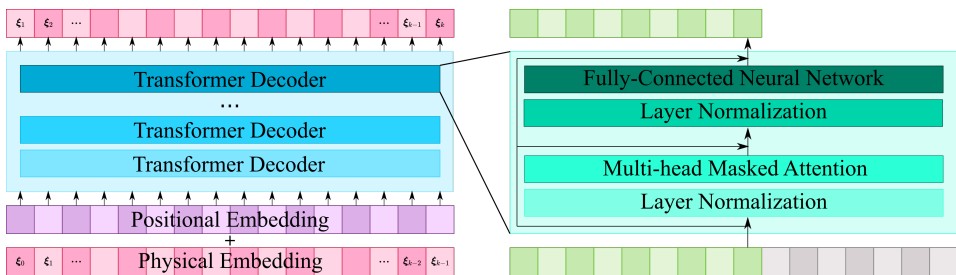

Figure 2: The transformer decoder model used for the prediction of physical dynamics.

Our model follows the GPT-2 architecture based on the implementation in the Hugging Face transformer repository (Wolf et al., 2019), but is significantly smaller in size than these modern NLP transformers. This model consists of a stack of transformer decoder layers that use masked attention, as depicted in Fig. 2. The inputs to the transformer are the embedded representation of the physical system with the sinusoidal positional encoding proposed in the original transformer (Vaswani et al., 2017). To train the model, consider a data set of $D$ embedded i.i.d. time-series $\mathcal{D} = \left\{\Xi^i\right\}_{i=1}^{D}$ for which we can use the standard time-series Markov model (language modeling) log-likelihood:

$$L_{\mathcal{D}} = \sum_i^D \sum_j^T - \log p\left(\boldsymbol{\xi}_j^i | \boldsymbol{\xi}_{j-k}^i, \ldots, \boldsymbol{\xi}_{j-1}^i, \boldsymbol{\theta}\right). \tag{2}$$

$k$ is the context window and $\boldsymbol{\theta}$ are the model's parameters. Contrary to the standard NLP approach which poses the likelihood as a softmax over a dictionary of tokens, the likelihood here is taken as a standard Gaussian between the transformer's prediction and the target embedded value resulting in a $L_2$ loss. This is due to the fact that the solution to most physical systems cannot be condensed to a discrete finite set making tokenization into a finite dictionary not possible and thus a softmax approach not applicable. Training is the standard auto-regressive method used in GPT (Radford et al., 2018), as opposed to the word masking (Devlin et al., 2018), constrained to the embedded space. The physical states, $\phi_i$, have the potential to be very high dimensional thus training the transformer in the lower-dimensional embedded space can significantly lower training costs.

## 2.2 EMBEDDING MODEL

The second major component of the machine learning methodology is the embedding model responsible for projecting the discretized physical state space into a 1D vector representation. In NLP, the standard approach is to tokenize then embed a finite vocabulary of words, syllables or characters using methods such as n-gram models, Byte Pair Encoding (Gage, 1994), Word2Vec (Mikolov et al., 2013a;b), GloVe (Pennington et al., 2014), etc. These methods allow language to be represented by a series of 1D vectors that serve as the input to the transformer. Clearly a finite tokenization and such NLP embeddings are not directly applicable to a physical system, thus we propose our own embedding method designed specifically for dynamical systems. Consider learning the generalized mapping between the system's state space and embedded space: $\mathcal{F} : \mathbb{R}^{n \times d} \to \mathbb{R}^e$ and $\mathcal{G} : \mathbb{R}^e \to \mathbb{R}^{n \times d}$. Naturally, multiple approaches can be used especially if the dimensionality of the embedded space is less than that of the state-space but this is not always the case.

The primary approach that we will propose is a Koopman observable embedding which is a technique that can be applied universally to all dynamical systems. Considering the discrete time form of the dynamical system in Eq. 1, the evolution of the state variables can be abstracted by $\phi_{i+1} = \mathbb{F}(\phi_i)$ for which $\mathbb{F}$ is the dynamic map from one time-step to the next. The foundation of Koopman theory states that for any dynamical system of this form, there exists an infinite set of state

observables, $g(\phi_i)$, that evolve linearly in time such that:

$$\mathcal{K}g(\phi_i) \triangleq g \circ \mathbb{F}(\phi_i),\tag{3}$$

where $\mathcal{K}$ is known as the Koopman operator (Koopman, 1931) which is time-invariant. Namely, the Koopman observables can evolve in time through continual matrix multiplication of with the Koopman operator:

$$g(\phi_{i+1}) = \mathcal{K}g(\phi_i), \quad g(\phi_{i+2}) = \mathcal{K}^2 g(\phi_i), \quad g(\phi_{i+3}) = \mathcal{K}^3 g(\phi_i), \ldots\tag{4}$$

in which $\mathcal{K}^n$ denotes $n$ matrix products (*e.g.* $\mathcal{K}^3 = \mathcal{K} \cdot \mathcal{K} \cdot \mathcal{K}$). Modeling the dynamics of a system through the linear Koopman space can be attractive due to its simplification of the dynamics but also the potential physical insights it brings along with it. Spectral analysis of the Koopman operator can reveal fundamental dynamical modes that drive the system's evolution in time.

However, these observables, $g(\phi_i)$, are typically unknown and are *theoretically* infinite dimensional. Thus Koopman theory can be viewed as a trade off between lifting the state space into observable space with more complex states but with simpler dynamics. For practical application, the Koopman operator and observables are finitely approximated. In recent years, various machine learning based approaches have been proposed to learn both the Koopman operator and observables approximated in a finite space for modeling, control and dynamical mode analysis (Takeishi et al., 2017; Li et al., 2017; Lusch et al., 2018; Korda & Mezić, 2018; Otto & Rowley, 2019; Korda et al., 2020; Mezic, 2020). While deep learning methods have enabled greater success with discovering Koopman observables and operators (Lusch et al., 2018; Otto & Rowley, 2019), applications have still been limited to fairly simple systems and do not work for long time prediction. This is likely due to the approximation of the finite-dimensional Koopman observables, limited data and complete dependence on the discovered Koopman operator $\mathcal{K}$ to model the dynamics. Suggesting the prediction of a system through a single linear transform clearly has significant limitations and is fundamentally a naive approach from a machine learning perspective.

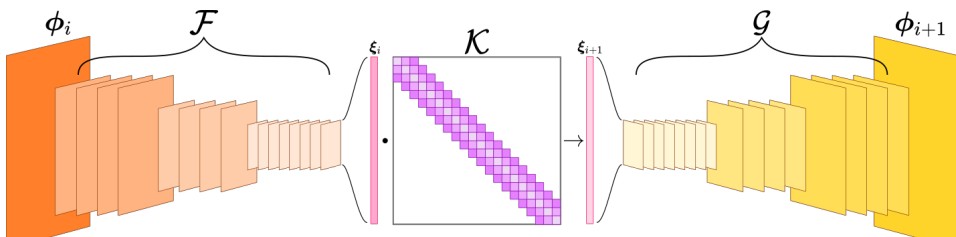

Figure 3: Example of a Koopman embedding for a two-dimensional system using a convolution encoder-decoder model. The encoder model, $\mathcal{F}$, projects the physical states into the approximate Koopman observable embedding. The decoder model, $\mathcal{G}$ recovers the physical states from the embedding.

In this work, we propose using approximate Koopman observations as a methodology to develop embeddings for the transformer model such that $\mathcal{F}(\phi_i) \triangleq g(\phi_i)$. As seen in Fig. 3, the embedding model follows a standard encoder-decoder model with the middle latent variables being the Koopman observables. Embedding model architectures for each numerical example are provided in Appendix A. We introduce a learnable Koopman operator which takes the form of a banded matrix that is symmetrical about the diagonal to encourage the discovery of dominate dynamical modes rather than high-frequency oscillations (Lusch et al., 2018; Otto & Rowley, 2019). This learned Koopman operator is disposed of once training of the embedding model is complete. Given the data set of physical state time-series, $\mathcal{D}_\Phi = \{\Phi^i\}_i^D$, the Koopman embedding model is trained with the following loss:

$$\mathcal{L}_{\mathcal{D}_\Phi} = \sum_{i=1}^{D} \sum_{j=0}^{T} \lambda_0 \underbrace{MSE\left(\phi_j^i, \mathcal{G} \circ \mathcal{F}\left(\phi_j^i\right)\right)}_{Reconstruction} + \lambda_1 \underbrace{MSE\left(\phi_j^i, \mathcal{G} \circ \mathcal{K}^j \mathcal{F}\left(\phi_0^i\right)\right)}_{Dynamics} + \lambda_2 \underbrace{\|\mathcal{K}\|_2^2}_{Decay}.\tag{5}$$

This loss function consists of three components: the first is a reconstruction loss which ensures a consistent mapping to and from the embedded representation. The second is the Koopman dynamics

loss which pushes $\boldsymbol{\xi}_j$ to follow linear dynamics resulting in time-steps of similar dynamics to have similar embeddings. The last term decays the Koopman operator's parameters to help force the model to discover lower-dimensional dynamical modes.

In reference to traditional NLP embeddings, we believe our Koopman observable embedding has a motivation similar to Word2Vec (Mikolov et al., 2013a) as well as more recent embedding methods such as context2vec (Melamud et al., 2016), ELMo (Peters et al., 2018), etc. Namely, these methods are based on the principle of context: words that are contextually related to each other have a similar embedding. The Koopman embedding has a similar objective encouraging physical realizations containing similar *time-invariant dynamical modes* to also have similar embeddings. Hence, our goal with the embedding model is to not find the true Koopman observables or operator, but rather leverage Koopman to enforce physical context.

## 3   CHAOTIC DYNAMICS

As a foundational numerical example to rigorously compare the proposed model to other classical machine learning techniques, we will first look at surrogate modeling of the Lorenz system governed by:

$$\frac{dx}{dt} = \sigma\left(y - x\right), \quad \frac{dy}{dt} = x\left(\rho - z\right) - y, \quad \frac{dz}{dt} = xy - \beta z. \tag{6}$$

We use the classical parameters of $\rho = 28, \sigma = 10, \beta = 8/3$. For this numerical example, we wish to develop a surrogate model for predicting the Lorenz system given a random initial state $x_0 \sim \mathcal{U}(-20, 20)$, $y_0 \sim \mathcal{U}(-20, 20)$ and $z_0 \sim \mathcal{U}(10, 40)$. In other words, we wish to surrogate model various initial value problems for this system of ODEs. The Lorenz system is used here because of its well known chaotic dynamics which make it extremely sensitive to numerical perturbations and thus an excellent benchmark for assessing a machine learning model's accuracy.

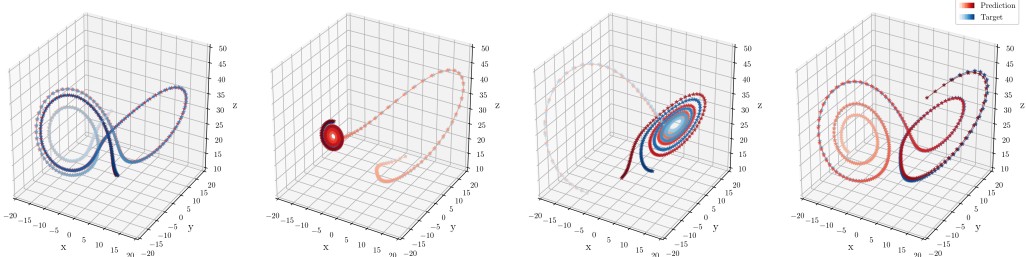

Figure 4: Four test case predictions using the transformer model for 320 time-steps.

A total of four alternative machine learning models are implemented: a fully-connected auto-regressive model, a fully-connected LSTM model, a deep neural network Koopman model (Lusch et al., 2018; Otto & Rowley, 2019) and lastly an echo-state model (Chattopadhyay et al., 2019; Lukoševičius, 2012). All of these models have been proposed in past literature for predicting many dynamical systems. Each are provided the same training, validation and testing data sets containing 2048, 64 and 256 time-series solved using a numerical solver, respectively. The training data set contains time-series of 256 time-steps while the validation and testing data sets have 1024 time-steps. Each model is allowed to train for 500 epochs if applicable. The proposed transformer and embedding model are trained for 200 and 300 epochs, respectively, with an embedding dimension of 32. The embedding model is a simple fully-connected encoder-decoder model, $\mathcal{F} : \mathbb{R}^3 \to \mathbb{R}^{32}; \mathcal{G} : \mathbb{R}^{32} \to \mathbb{R}^3$, which is trained first. The transformer is then trained with a context length of 64 with 4 transformer decoder layers.

We plot four separate test cases in Fig. 4 for which only the initial state is provided and the transformer model predicts 320 time-steps. Several test predictions for alternative models are provided in Appendix B. In general, we can see the transformer model is able to yield extremely accurate predictions even beyond its context length. Additionally, we plot the Lorenz solution for 25k time-steps from a numerical solver and predicted from transformer model in Fig. 5a. Note both have the same structure, which qualitatively indicates that the transformer indeed maintains physical dynamics.

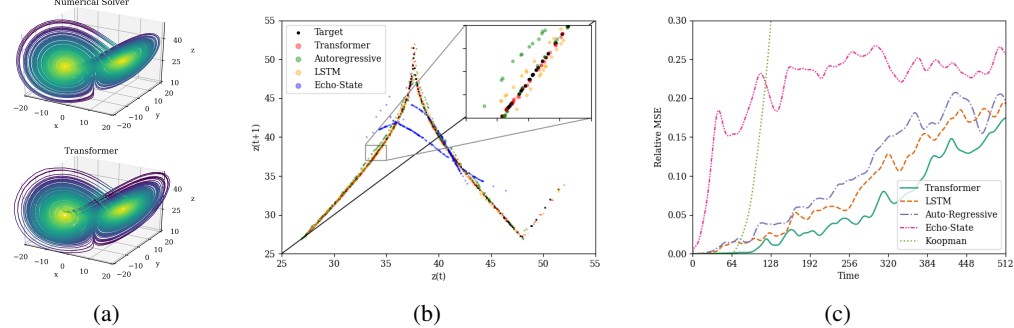

(a)            (b)            (c)

Figure 5: (a) Lorenz solution of 25k time-steps. (b) The Lorenz map produced by each model. (c) The test relative mean-squared-error (MSE) with respect to time.

Table 1: Test set relative mean-squared-error (MSE) for surrogate modeling the Lorenz system at several time-step intervals.

| | | Relative MSE | | |
|---|---|---|---|---|
| Model | Parameters | [0-64) | [64-128) | [128-192) |
| Transformer | 36k/54k[†] | 0.0003 | **0.0060** | **0.0221** |
| LSTM | 103k | 0.0041 | 0.0175 | 0.0369 |
| Autoregressive | 92k | 0.0057 | 0.0253 | 0.0485 |
| Echo State | 7.5k/6.3m[‡] | 0.1026 | 0.1917 | 0.2209 |
| Koopman | 108k | **0.0001** | 0.0962 | 2.0315 |

[†] Learnable parameters for the embedding/transformer model.
[‡] Learnable output parameters/fixed input and reservoir parameters.

Table 2: Test set relative mean-squared-error (MSE) for surrogate modeling the Lorenz system at several time-step intervals with noisy data.

| | Relative MSE 1% Noise | | | Relative MSE 5% Noise | | |
|---|---|---|---|---|---|---|
| Model | [0-64) | [64-128) | [128-192) | [0-64) | [64-128) | [128-192) |
| Transformer | **0.0021** | **0.0216** | **0.0429** | 0.0210 | 0.0759 | **0.1292** |
| LSTM | 0.0045 | 0.0218 | 0.0437 | 0.0212 | **0.0758** | 0.1324 |
| Autoregressive | 0.0114 | 0.0417 | 0.0901 | 0.0760 | 0.2060 | 0.2065 |
| Echo State | 0.0859 | 0.1686 | 0.2102 | 0.1000 | 0.1581 | 0.2051 |
| Koopman | 0.0047 | 0.1192 | 0.1597 | **0.0200** | 0.0787 | 0.1906 |

The proposed transformer and alternative models' relative mean squared errors for the test set are plotted in Fig. 5c as a function of time and listed in Table 1 segmented into several intervals based on the transformer's context length. We can see that the transformer model is able to achieve the best performance using the least number of parameters with the exception of the Koopman only model for early times. However, the Koopman model is unstable for long time predictions rendering it useless for surrogate predictions. To quantify accuracy of the chaotic dynamics of each model, the Lorenz map is plotted in Fig. 5b which is a well-defined relation between successive $z$ local maximas despite the Lorenz's chaotic nature. Calculated using 25k time-step predictions from each model, again we can see that the transformer model agrees the best with the numerical solver indicating that it has learned the best physical dynamics of all the tested models.

Additionally we test the transformer's sensitivity to contaminated data by adding white noise to the training observations scaled by the magnitude of the state variables. Each model tested with clean data is retrained with data perturbed by 1% and 5% noise. The effects of these two different noise levels is qualitatively illustrated in Appendix B. The errors are listed in Table 2. In general, we can indeed see that the transformer can still perform adequately with noisy data by being the best performing model for 1% noise and still being competitive, particularly at later time-steps, with 5% noise.

## 4    2D FLUID DYNAMICS

The next dynamical system we will test is transient 2D fluid flow governed by the Navier–Stokes equations:

$$\frac{\partial u_i}{\partial t} + u_j \frac{\partial u_i}{\partial x_j} = -\frac{1}{\rho}\frac{\partial p}{\partial x_i} + \nu \frac{\partial u_i}{\partial x_j^2}, \tag{7}$$

in which $u_i$ and $p$ are the velocity and pressure, respectively. $\nu$ is the viscosity of the fluid. We consider modeling the classical problem of flow around a cylinder at various Reynolds number defined by $Re = u_{in}d/\nu$ in which $u_{in} = 1$ and $d = 2$ are the inlet velocity and cylinder diameter, respectively. In this work, we choose to develop a surrogate model to predict the solution at any Reynolds number between $Re \sim \mathcal{U}(100, 750)$. This problem is a fairly classical flow to investigate in scientific machine learning with various levels of difficulty (Lee & You, 2017; Morton et al., 2018; Lusch et al., 2018; Han et al., 2019; Xu & Duraisamy, 2020; Geneva & Zabaras, 2020b). Here we choose one of the more difficult forms: model the flow starting at a steady state flow field at $t = 0$. Meaning the model is provided zero information on the structure of the cylinder wake during testing.

Training, validation and testing data is obtained using the OpenFOAM simulator (Jasak et al., 2007), from which a rectangular structured sub-domain is sampled centered around the cylinder's wake. Given that the flow is two-dimensional, our model will predict the $x$-velocity, $y$-velocity and pressure fields, $\{\boldsymbol{u}_x, \boldsymbol{u}_y, \boldsymbol{p}\} \in \mathbb{R}^{3 \times 128 \times 64}$. The training, validation and test data sets consist of 27, 6 and 7 fluid flow simulations respectively with 400 time-steps each at a physical time-step size of $\Delta t = 0.5$. As a base line model, we train a convolutional encoder-decoder model with a stack of three convolutional LSTMs (Xingjian et al., 2015) in the center. We note that convolutional LSTMs have been used extensively in recent scientific machine learning literature for modeling various physical systems (Han et al., 2019; Wiewel et al., 2019; Tang et al., 2020; Geneva & Zabaras, 2020b), thus can be considered a state-of-the-art approach. The convolutional LSTM model is trained for 500 epochs.

Additionally, three different embedding methods are implemented: the first is the proposed Koopman observable embedding using a convolutional auto–encoder. This model encodes the fluid $x$-velocity, $y$-velocity, pressure and viscosity fields to an embedded dimension of 128, $\mathcal{F} : \mathbb{R}^{4 \times 128 \times 64} \to \mathbb{R}^{128}; \mathcal{G} : \mathbb{R}^{128} \to \mathbb{R}^{3 \times 128 \times 64}$. The second embedding method is using the same convolutional auto–encoder model but without the enforcement of Koopman dynamics on the embedded variables. The last embedding method tested was principal component analysis (PCA), a classical dimensionality reduction benchmark method for all fields of science. For each embedding method an identical transformer model with a context length of 128 with 4 transformer decoder layers is trained. Similar to the previous example, the embedding models are trained for 300 epochs when applicable and the transformer is trained for 200.

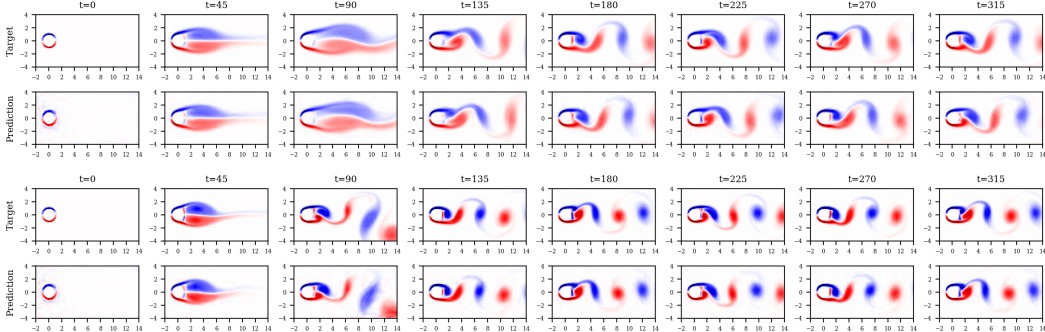

Figure 6: Vorticity, $\boldsymbol{\omega} = \nabla_x \boldsymbol{u}_y - \nabla_y \boldsymbol{u}_x$, of two test case predictions using the proposed transformer with Koopman embeddings at Reynolds numbers 233 (top) and 633 (bottom).

Each trained model is tested on the test set by providing the initial laminar state at $t = 0$ with the fluid viscosity and allowing the model to predict 400 time-steps into the future. Two test predictions using the proposed transformer model with Koopman embeddings are plotted in Fig. 6 in which the predicted vorticity fields are in good agreement with the true solution. The test set relative mean

square error for each output field for each model is plotted in Fig. 7. The errors of each field over the entire time-series are listed in Table 3. Additional results are provided in Appendix C.

For all alternative models, a rapid error increase can be seen between $t = [50, 100]$ which is due to the transition from the laminar flow into vortex shedding. This error then plateaus since each model is able to produce stable vortex shedding, as illustrated in Figs. 15 & 16 in Appendix C, however the instantaneous states predicted from these models deviate from the true numerical solution. It is clear that only the transformer model with Koopman embeddings can correctly predict these dynamics demonstrating the benefits of enforcing Koopman dynamics on the embedded representations. Compared to the widely used ConvLSTM model, the proposed transformer offers far more reliable predictions with less learnable parameters.

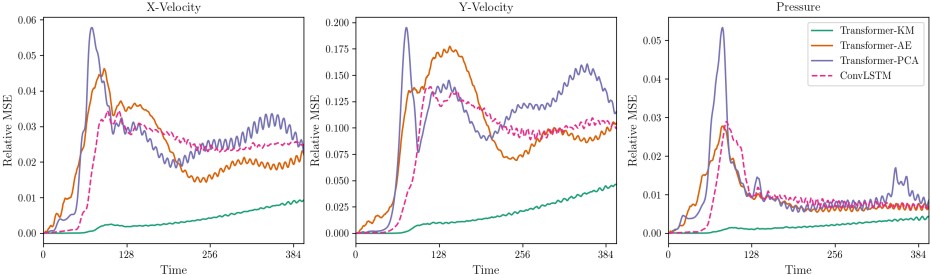

Figure 7: Test set relative mean-squared-error (MSE) of the base-line convolutional LSTM model (ConvLSTM) and transformers with Koopman (KM), auto-encoder (AE) and PCA embedding methods over time.

Table 3: Test set relative mean-squared-error (MSE) of each output field for surrogate modeling 2D fluid flow past a cylinder. Models listed include the convolutional LSTM model (ConvLSTM) and transformers with Koopman (KM), auto-encoder (AE) and PCA embedding methods.

| Model | Parameters | Relative MSE $[0 - 400]$ | | |
|---|---|---|---|---|
| | | $\boldsymbol{u}_x$ | $\boldsymbol{u}_y$ | $\boldsymbol{p}$ |
| Transformer-KM | 224k/628k[†] | **0.0036** | **0.0175** | **0.0018** |
| Transformer-AE | 224k/628k[†] | 0.0226 | 0.09606 | 0.0088 |
| Transformer-PCA | 3.1m/628k[†] | 0.0251 | 0.1062 | 0.0108 |
| ConvLSTM | 934k | 0.0220 | 0.0862 | 0.0086 |

[†] Learnable parameters for the embedding/ transformer model.

## 5 3D REACTION–DIFFUSION DYNAMICS

The final numerical example to demonstrate the proposed transformer model is a 3D reaction–diffusion system governed by the Gray–Scott model:

$$\frac{\partial u}{\partial t} = r_u \frac{\partial u}{\partial x_i^2} - uv^2 + f(1 - u), \quad \frac{\partial v}{\partial t} = r_v \frac{\partial v}{\partial x_i^2} + vu^2 - (f + k)v, \quad (8)$$

in which $u$ and $v$ are the concentration of two species, $r_u$ and $r_v$ are their respective diffusion rates, $k$ is the kill rate and $f$ is the feed rate. This is a classical system of particular application to chemical processes as it models the following chemical reaction: $U + 2V \to 3V; V \to P$. For a set region of feed and kill rates, this seemingly simple system can yield a wide range of complex dynamics (Pearson, 1993; Lee et al., 1993). Hence, under the right settings, this system is an excellent case study to push the proposed methodology to its predictive limits. In this work, we will use the parameters: $r_u = 0.2$, $r_v = 0.1$, $k = 0.055$ and $f = 0.025$ which results in a complex dynamical reaction seen in Fig. 8. Akin to the first numerical example, the initial condition of this system is stochastic such that the system is seeded with 3 randomly placed perturbations within the periodic domain.

Training, validation and testing data are obtained from a Runge-Kutta finite difference simulation on a structured grid, $\{\boldsymbol{u}, \boldsymbol{v}\} \in \mathbb{R}^{2 \times 32 \times 32 \times 32}$. The training, validation and test data sets consist

of 512, 16 and 56 time-series, respectively, with 200 time-steps each at a physical time-step size of $\Delta t = 5$. A 3D convolutional encoder-decoder is used to embed the two species into a 512 embedding dimension, $\mathcal{F} : \mathbb{R}^{2 \times 32 \times 32 \times 32} \to \mathbb{R}^{512}; \mathcal{G} : \mathbb{R}^{512} \to \mathbb{R}^{2 \times 32 \times 32 \times 32}$. Transformer models with varying depth are trained all with a context length of 128. All other model and training parameters for the transformer models are consistent. A test prediction using the transformer model is shown in Fig. 8 and the errors for each trained transformer are listed in Table 4. Despite this system having complex dynamics in 3D space, the transformer is able to produce acceptable predictions with very similar structures as the numerical solver. To increase the model's predictive accuracy we believe the limitation here is not in the transformer, but rather the number of training data and the inaccuracies of the embedding model due to the dimensionality reduction needed. This is supported by the fact that increasing the transformer's depth does not yield considerable improvements for the test errors in Table 4. Additional, results are provided in Appendix D.

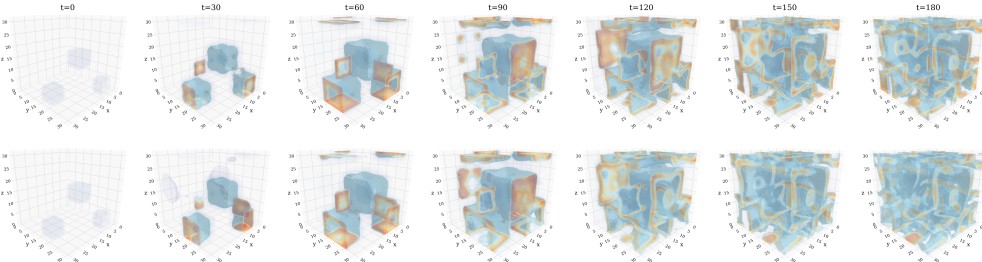

Figure 8: $u$ volume plots of the target (top) and transformer prediction (bottom) for a test case of the Gray–Scott system. Isosurfaces displayed span the range $u = [0.3, 0.5]$ to show the inner structure.

Table 4: Test set relative mean-squared-error (MSE) for surrogate modeling 3D Gray-Scott system.

| Model | Layers | Parameters | Relative MSE $[0 - 200]$ | |
| | | | $u$ | $v$ |
|---|---|---|---|---|
| Transformer | 2 | 6.2m/6.6m[†] | 0.0159 | 0.0120 |
| Transformer | 4 | 6.2m/12.9m[†] | 0.0154 | 0.0130 |
| Transformer | 8 | 6.2m/25.5m[†] | **0.0125** | **0.0101** |

[†] Learnable parameters for the embedding/ transformer model.

## 6 CONCLUSION

While transformers and self-attention models have been established as a powerful framework for NLP tasks, the adoption of such methodologies has yet to fully permute other fields. In this work, we have demonstrated the potential transformers have for modeling dynamics of physical phenomena. The transformer architecture with self-attention allows the model to learn longer and more complex temporal dependencies than alternative machine learning methods, which is benign for many physical systems that exhibit multi-scale dynamics. This can be directly attributed to the transformers ability to draw information from many past time-steps efficiently through self-attention rather than just the most recent time-step which occurs in auto-regressive, Euler-based and standard RNN models.

The key challenge of using the transformer model is identifying appropriate embeddings to represent the physical state of the system, for which we propose the use of Koopman dynamics to enforce dynamical context. Using the proposed methods, our transformer model can outperform alternative models widely seen in recent scientific machine learning literature. The investigation of unsupervised pre-training of such transformer models as well as gaining a better understanding of what attention mechanisms imply for physical systems will be the subjects of works in the near future.

### ACKNOWLEDGMENTS

The authors acknowledge computing resources provided by the AFOSR Office of Scientific Research through the DURIP program.

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

## A  KOOPMAN EMBEDDING MODELS

The architectures for the Koopman embedding models are provided. In general, each is a classical auto-encoder architecture using standard machine learning methodologies. For complete details we refer the reader to the code.

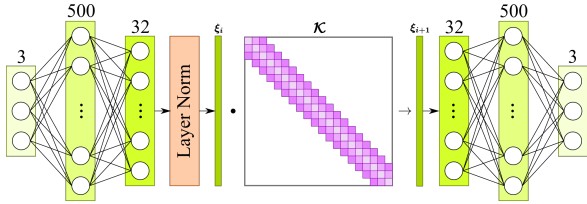

Figure 9: Fully-connected embedding network with ReLU activation functions for the Lorenz system.

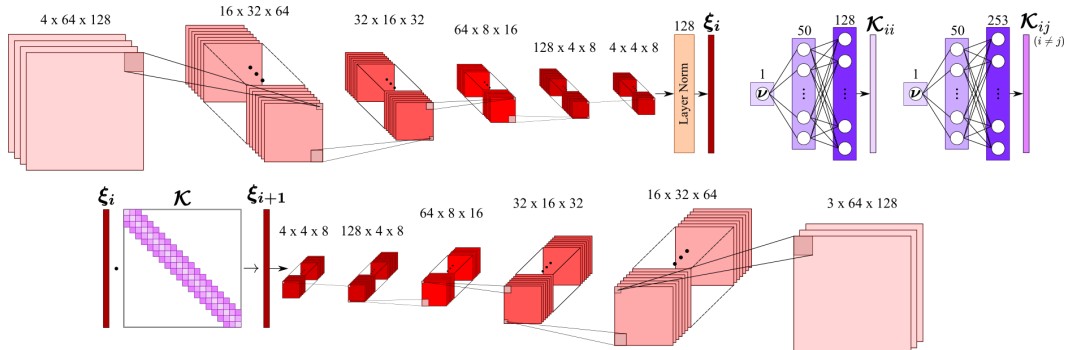

Figure 10: 2D convolutional embedding network with ReLU activation functions for the flow around a cylinder system consisting of 5 convolutional encoding/decoding layers. Each convolutional operator has a kernel size of $(3, 3)$. In the decoder, the feature maps are up-sampled before applying a standard convolution. Additionally, two auxiliary fully-connected networks are used to predict the diagonal and off-diagonal elements of the Koopman operator for each viscosity $\nu$.

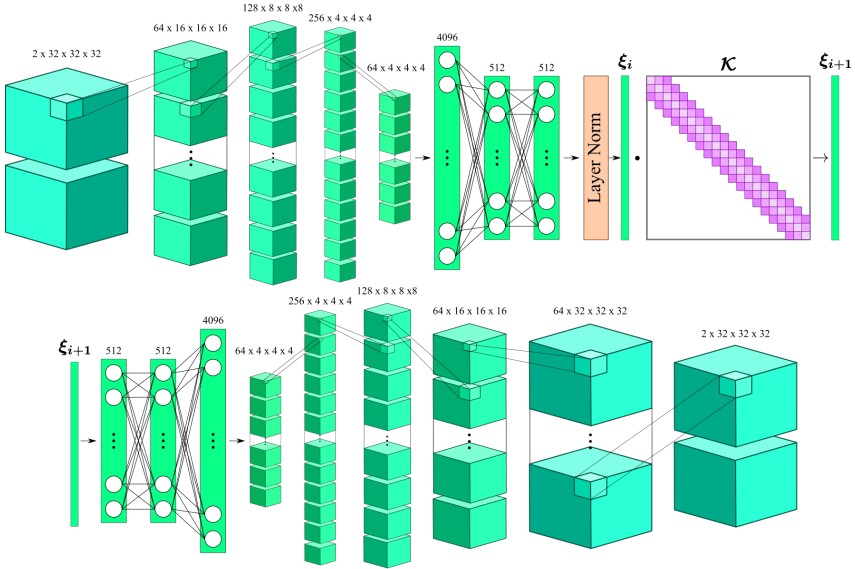

Figure 11: 3D convolutional embedding network with leaky ReLU activation functions for the Gray-Scott system. Batch-normalization used between each of the convolutional layers. In the decoder, the feature maps are up-sampled before applying a standard 3D convolution.

# B   LORENZ SUPPLEMENTARY RESULTS

Predictions for all of the tested models for three test cases are plotted in Fig. 12 in which we can see the transformer model is consistently accurate within the plotted time frame. Additional test predictions for the Lorenz system using the transformer model are shown in Fig. 13. Lastly, in Fig. 14 we illustrate the two different noise levels used to contaminate the training data for the results listed in Table 2.

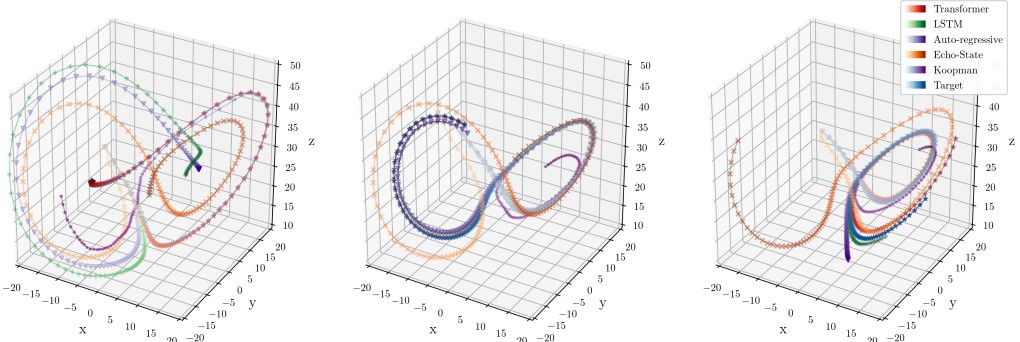

Figure 12: Three Lorenz test case predictions using each tested model for 128 time-steps.

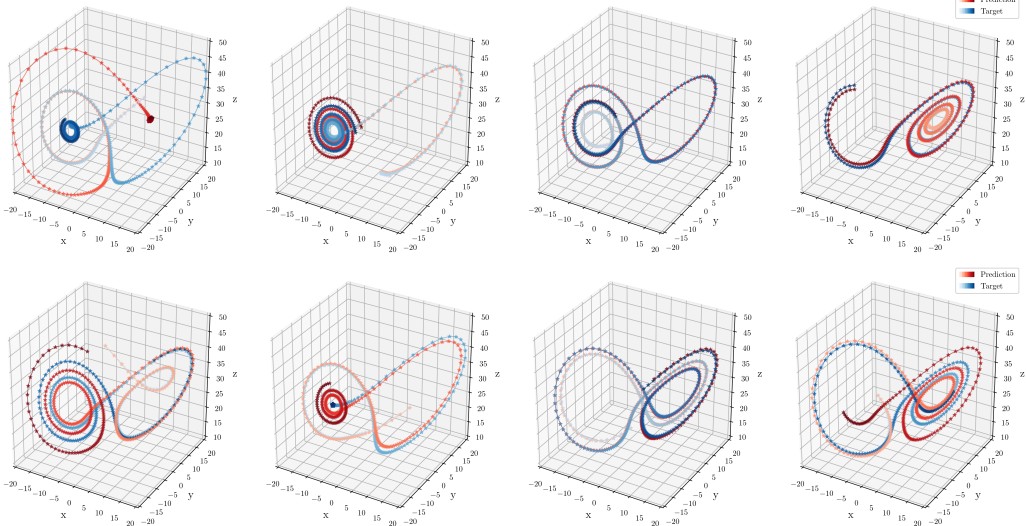

Figure 13: Lorenz test case predictions using the transformer model for 320 time-steps.

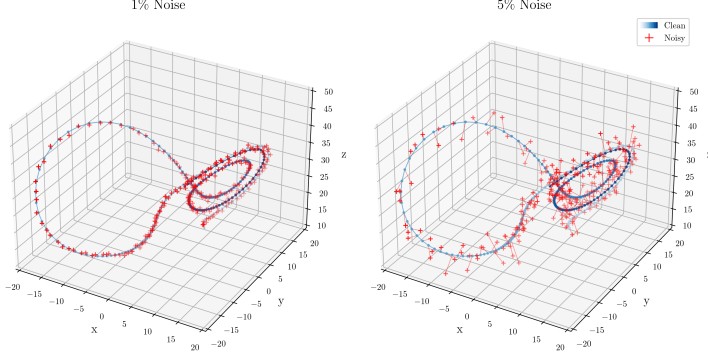

Figure 14: Comparison between the clean and contaminated (noisy) training data.

## C    CYLINDER SUPPLEMENTARY RESULTS

Prediction fields of all the tested models for a single test case are plotted in Figs. 15 & 16. While all models are able to perform adequately with qualitatively good results, the transformer model with Koopman embeddings (Transformer-KM) outperforms alternatives.

The evolution of the flow field projected onto the dominate eigenvectors of the learned Koopman operator for two Reynolds numbers are plotted in Fig. 17. This reflects the dynamical modes that were learned by the embedding model to impose physical "context" on the embeddings. Given that the eigenvectors are complex, we plot both the magnitude, $|\psi|$, and angle, $\angle\psi$. For both Reynolds numbers it is easy to see the initial transition region, $t < 100$, before the system enters the periodic vortex shedding. Once in the periodic region, we can see that the higher Reynolds number has a higher frequency which reflects the increased vortex shedding speed.

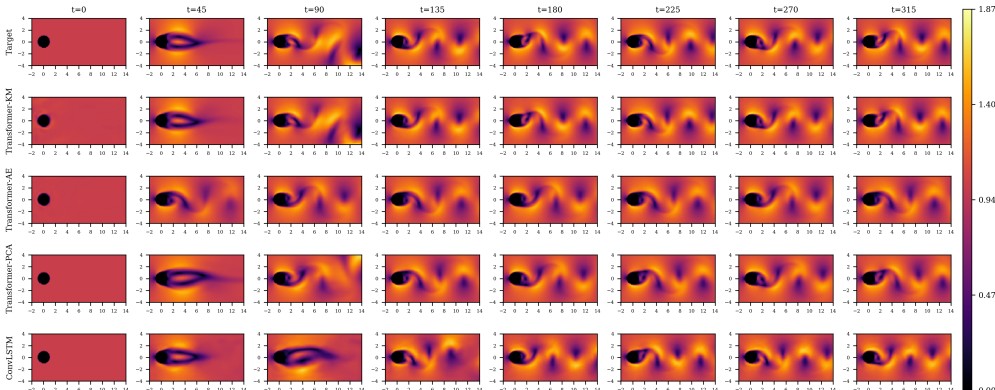

Figure 15: Velocity magnitude predictions of a test case at $Re = 633$ using each tested model.

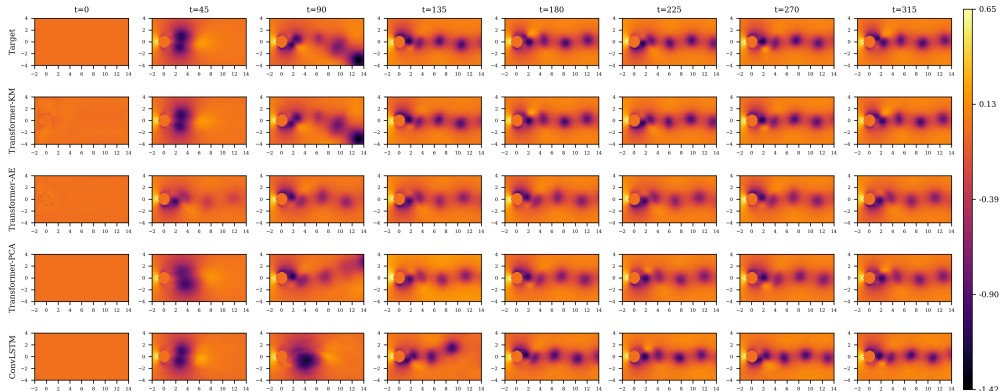

Figure 16: Pressure predictions of a test case at $Re = 633$ using each tested model.

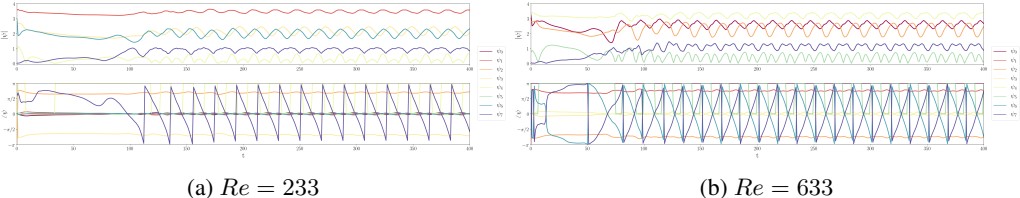

(a) $Re = 233$                           (b) $Re = 633$

Figure 17: The dynamics of the fluid flow around a cylinder projected onto the 8 most dominate eigenvectors of the learned Koopman operator, $\mathcal{K}$, in the embedding model.

## D  GRAY-SCOTT SUPPLEMENTARY RESULTS

Volume plots of two test cases for both species are provided in Figs. 18 & 19. Additionally, to better visualize the accuracy of the trained transformer model, contour plots for three test cases are also provided in Fig. 20. Both the volume and contour plots illustrate the complex and unique structures that arise in this system for each initial condition.

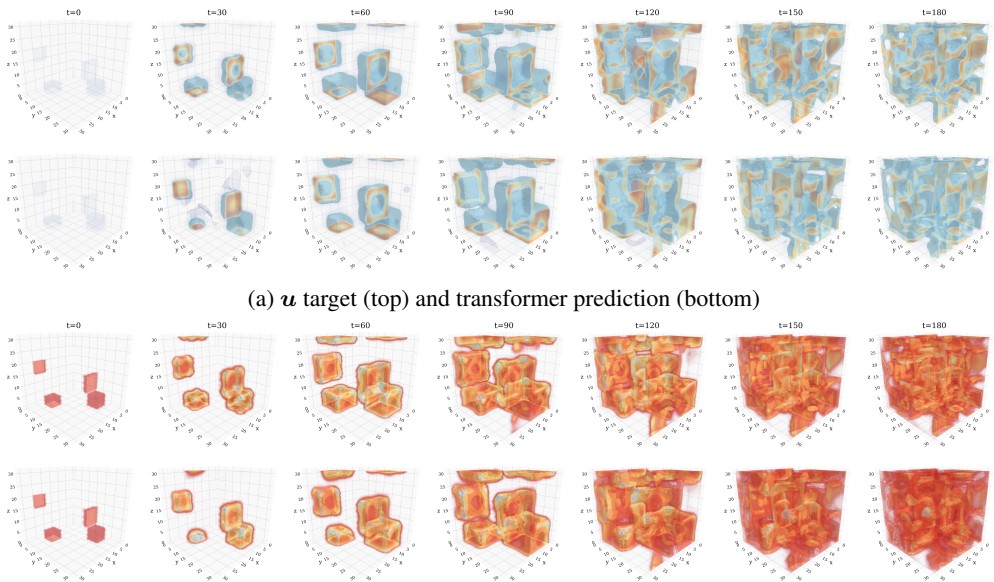

(a) $u$ target (top) and transformer prediction (bottom)

(b) $v$ target (top) and transformer prediction (bottom)

Figure 18: Test case volume plots for the Gray–Scott system. Isosurfaces displayed span the range $u, v = [0.3, 0.5]$ to show the inner structure.

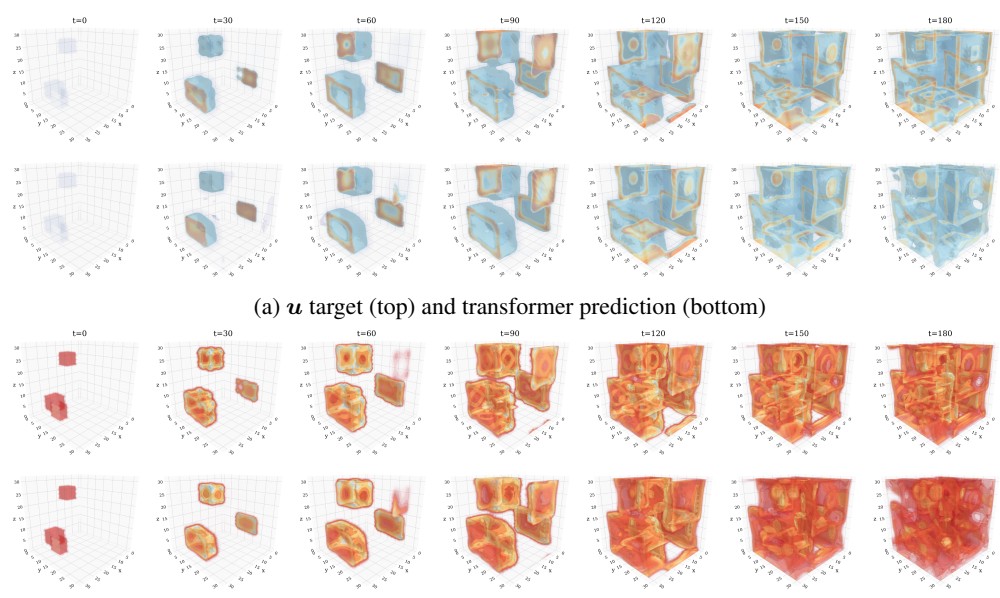

(a) $u$ target (top) and transformer prediction (bottom)

(b) $v$ target (top) and transformer prediction (bottom)

Figure 19: Test case volume plots for the Gray–Scott system. Isosurfaces displayed span the range $u, v = [0.3, 0.5]$ to show the inner structure.

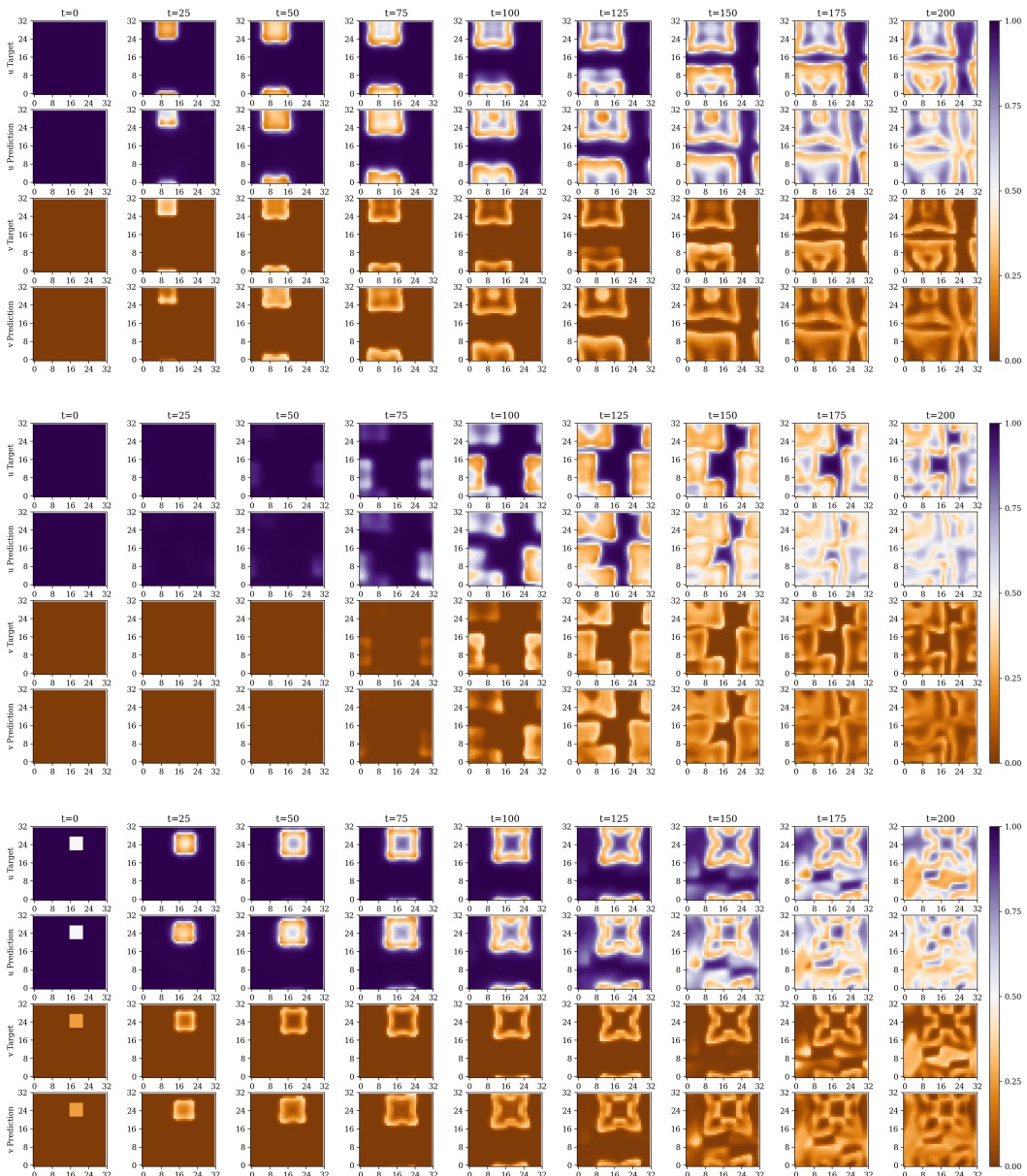

Figure 20: $x - y$ plane contour plots of three Gray–Scott test cases sliced at $z = 16$.

