# OpenReview forum: "Transformers for Modeling Physical Systems"
_ICLR.cc/2021/Conference — Reject_

### Official Review · AnonReviewer1 · 2020-10-20
**Clear and well executed paper that currently misses significant experimental evaluation**

**Rating:** 6
**Confidence:** 4

**Review:**

### Summary:
The paper proposes to use transformers for modelling dynamical systems. The transformer is combined with a linear dynamical system to enforce Koopman features and is trained using the reconstruction and prediction loss. Finally, the proposed algorithm is applied to the different tasks with 1, 2 & 3 dimensions. On each simulated task the proposed algorithm marginally outperforms sufficient baselines.

### Clarity & Style:
The paper is clearly written and understandable. The figures could be drastically improved by optimizing the white space and axes spacing, e.g. figure 5. Furthermore, having videos of the solver and network predictions would be preferable. It's a bit surprising that the authors don't cite https://arxiv.org/pdf/2002.09405.pdf, which would also be an interesting but not necessarily required baseline.


### Quality, Originality & Significance:
While the paper is clearly written and well executed in terms of experiments and baselines, the paper is missing originality and significance. The paper does not introduce new perspective or demonstrate a previously unaccomplished task. It simply applies a straight-forward method and demonstrates that it works marginally better compared to existing methods. Especially the usefulness of the model is questionable. The Lorenz system is a **three** parameter dynamical system and is approximated with a transformer with **54,000** parameters! What is the advantage of the transformer network compared to the analytical solvers? Is it faster and if yes how much more energy does it consume compared to the solver? Why is it better compared to a look-up table with 54,000 values, which does not require tuning network hyper-parameters? Does an analytic version of the Koopman operator exist for the Lorenz system? There might be a good chance as the Lorenz system is a polynomial dynamic system. How many parameters does this have? If so how do the learned features compare to the analytic features?

Furthermore, the paper lacks to address the real challenges of learning dynamical systems (*in my opinion*). For clean data (or data with clean Gaussian noise) and a single coherent timescale most methods perform reasonably well. IMO the main problems are:

* Messy data from real systems including, non-Gaussian noise, discretization errors, time delays. Can such an over-parameterized transformer work for such data and recover the underlying structure or does it overfit?
* Dynamical systems with multiple space & time-scales. Is the transformer model able to learn a good dynamical system with multiple time-scales?

To increase the significance of the paper the authors would need to address one of these two questions.


###  Conclusion:
All in all, I think the paper is well written and the performed experiments are well executed. However, right now the paper misses significance to be published at ICLR. To improve the papers the authors would either have to address messy real-world data or dynamical systems with multiple time-scales. I am open for discussion, why this model is a significant advancement over prior methods, but right now I don't see that.

### **Post Discussion Comments:**
The authors provided videos of the task. While I still find that this paper misses clear benefits/use-case for the machine-learning community. I still don't clearly understand why one would need 54,000 parameters for a  3 parameter system. One could store a really big table of this one 1d system for this amount of parameters. However, the paper is well executed and clearly written in regards of technical aspects. The motivation remains doubtful for me. Due to the execution and clear writing, i increased my score to weak accept.

---

> ### Author Response · Authors · 2020-11-13
> **Anon Reviewer 1 Response (1)**
>
> **We address each part reviewer's comments below:**
>
> *Summary: The paper proposes to use transformers for modelling dynamical systems. The transformer is combined with a linear dynamical system to enforce Koopman features and is trained using the reconstruction and prediction loss. Finally, the proposed algorithm is applied to the different tasks with 1, 2 & 3 dimensions. On each simulated task the proposed algorithm marginally outperforms sufficient baselines.*
>
> **The authors thank the reviewer for their comments and suggestions. Their effort to provide feedback for the submitted paper is greatly appreciated.**
>
> *Clarity & Style: The paper is clearly written and understandable. The figures could be drastically improved by optimizing the white space and axes spacing, e.g. figure 5. Furthermore, having videos of the solver and network predictions would be preferable.*
>
> **The authors assure the reviewer that videos for each of the numerical test cases have already been prepared and will be available through the github repositories website. Yet these have not been provided due to anonymity requirements. They will be added after the review process is complete through. Thank you.**
>
> *It's a bit surprising that the authors don't cite https://arxiv.org/pdf/2002.09405.pdf, which would also be an interesting but not necessarily required baseline.*
>
> **While such a graphical model for Lagrangian based simulations is not directly applicable to the numerical examples of interest, this is indeed a relevant work and has been cited. Perhaps  transformers/self-attention could also be used to increase the accuracy of the model in this reference. Thank you for bringing this to our attention.**
>
> *Quality, Originality & Significance: While the paper is clearly written and well executed in terms of experiments and baselines, the paper is missing originality and significance. The paper does not introduce new perspective or demonstrate a previously unaccomplished task. It simply applies a straight-forward method and demonstrates that it works marginally better compared to existing methods.*
>
> **The originality and significance of this paper lies in several key factors: the first is the proposition and clear demonstration of transformers/self-attention to predict physical dynamics. Additionally, we place an emphasis on the use of a physics-inspired Koopman theory embeddings that directly injects physical context into the transformer model. We pose a new perspective by suggesting that indeed language modeling can be viewed as a time-series problem and methods used in NLP can directly extend to physical systems. Our paper demonstrates the potential self-attention for time-integration of dynamics which is of great interest to many engineers, physicist, chemists, etc in both academia and industry.**
>
> *Especially the usefulness of the model is questionable. The Lorenz system is a three parameter dynamical system and is approximated with a transformer with 54,000 parameters! What is the advantage of the transformer network compared to the analytical solvers? Is it faster and if yes how much more energy does it consume compared to the solver? Why is it better compared to a look-up table with 54,000 values, which does not require tuning network hyper-parameters? Does an analytic version of the Koopman operator exist for the Lorenz system? There might be a good chance as the Lorenz system is a polynomial dynamic system. How many parameters does this have? If so how do the learned features compare to the analytic features?*
>
> **The Lorenz system is one of the most widely studied paradigms for understanding dynamics. The reason for this, as stated in the paper, is that the system is chaotic and highly sensitive to numerical perturbations. The goal of this particular numerical example is to not beat a numerical solver, but rather test the models' predictive accuracy to a classical numerical solver rigorously. The sensitivity of the Lorenz systems is the reason why it is a classical benchmark for dynamical models in the scientific machine learning community. The demonstration that the embedding methodology and the transformer work accurately for surrogate modeling the system is a significant milestone on its own.**
>
> **The following two numerical studies, flow around a cylinder and the Gray-Scott system, are both good examples of a practical surrogate. For example, the transformer is order of magnitudes faster than the finite volume simulation of the fluid flow, but this is not the intended focus of the paper.**

---

> > ### Author Response · Authors · 2020-11-13
> > **Anon Reviewer 1 Response (2)**
> >
> > **Our response to the reviewer's comments continue below:**
> >
> > *Furthermore, the paper lacks to address the real challenges of learning dynamical systems (in my opinion). For clean data (or data with clean Gaussian noise) and a single coherent timescale most methods perform reasonably well. IMO the main problems are:*
> >
> > *-- Messy data from real systems including, non-Gaussian noise, discretization errors, time delays. Can such an over-parameterized transformer work for such data and recover the underlying structure or does it overfit?*
> >
> > *-- Dynamical systems with multiple space & time-scales. Is the transformer model able to learn a good dynamical system with multiple time-scales?*
> >
> > *To increase the significance of the paper the authors would need to address one of these two questions.*
> >
> > **The authors are actively looking into performing additional noisy data tests. We will update the reviewer in the following days.**
> >
> > **Multi-scale dynamics do indeed exist in the  numerical examples considered. For the flow around the cylinder, for a single Reynolds number the time-scales between the initial laminar, transition and vortex shedding phases are all very different. This is illustrated by the eigen projections provided in Fig. 15 in the appendix. Additionally, since the model is indeed a surrogate for multiple Reynolds numbers, this implies the transformer must predict vortex shedding at different frequencies and thus multiple time-scales. Further more the Gray-Scott model exhibits multi-scale nature with initial reaction front propagating rapidly but then slowing significantly when encountering another. Hence, we believe that our examples do exhibit a selection of different multi-scale behavior sufficient to demonstrate our methods applicability.**
> >
> > *Conclusion: All in all, I think the paper is well written and the performed experiments are well executed. However, right now the paper misses significance to be published at ICLR. To improve the papers the authors would either have to address messy real-world data or dynamical systems with multiple time-scales. I am open for discussion, why this model is a significant advancement over prior methods, but right now I don't see that.*
> >
> > **We have added many details in the paper to make our paper generally more clear for the reader as well as make the contribution very explicit. Thank you.**

---

> > > ### Comment · AnonReviewer1 · 2020-11-14
> > > **Videos**
> > >
> > > _The authors assure the reviewer that videos for each of the numerical test cases have already been prepared and will be available through the github repositories website. Yet these have not been provided due to anonymity requirements._
> > >
> > > **Great release them through an anonymus site. Many other paper do it through google sites and do not break double blind in the process. Not releasing the videos due to anonymity problems is just a made up argument.**

---

> > > > ### Author Response · Authors · 2020-11-15
> > > > **Videos**
> > > >
> > > > We have uploaded the videos to the following google site for the review process, thank you for the suggestion.
> > > >
> > > > [https://sites.google.com/view/transformersphysx](https://sites.google.com/view/transformersphysx)

---

> > > > > ### Comment · AnonReviewer1 · 2020-11-24
> > > > > **Videos**
> > > > >
> > > > > Thanks for uploading the videos. I have adapted my review with post-rebuttal comments.

---

> > > ### Author Response · Authors · 2020-11-17
> > > **Anon Reviewer 1 Response (3)**
> > >
> > > *Furthermore, the paper lacks to address the real challenges of learning dynamical systems (in my opinion). For clean data (or data with clean Gaussian noise) and a single coherent timescale most methods perform reasonably well. IMO the main problems are:*
> > >
> > > *-- Messy data from real systems including, non-Gaussian noise, discretization errors, time delays. Can such an over-parameterized transformer work for such data and recover the underlying structure or does it overfit?*
> > >
> > > **We have completed noisy runs of each model tested for the Lorenz system at two different noise levels and have added a table of results with a short discussion in section 3. We selected to test the Lorenz system with noise due to its well known numerical sensitivity and because this is the system we had the most comparable models implemented for. White noise was used here since this is the most widely used and natural data contamination approach for surrogate and inversion models that use training data produced from numerical solvers. The results demonstrate the transformers ability to avoid over-fitting and remain competitively accurate with alternative models.**

---

### Official Review · AnonReviewer3 · 2020-10-28
**Nice contribution**

**Rating:** 7
**Confidence:** 3

**Review:**

Quality


- The experiments are clear and the results are easily understandable.
- In all three experiments, the paper achieves compelling results with low errors compared to baselines, especially for high number of time steps.



Clarity


- I find the mathematical notations in Section 2.2 could be more clear. I'm a bit confused about the infinite dimensionality of the state observables and how the Koopman operator handles it, whereas in the Figure 2 shown the Koopman operator K looks like a banded diagonal linear operator. It is not apparent why equation (3) leads to the construction of the encoder-decoder model shown. More explanations on the Koopman operator would be appreciated to make the paper more self-contained.

- The plots make the experiment results quite clear in terms of the model's capability to estimate the dynamics.

- Some minor typos such as "on going" which should be "ongoing"




Originality

- This work uses a Transformer model instead of recurrent networks to perform dynamical system simulation. The paper also proposes using an embedding model. Overall, the novelty is moderate.


Significance



High-level pros and cons

Pros

- Modeling dynamical systems is an interesting deep learning application.
- Good results compared to baselines in all experiments shown.

Cons

The clarity can be improved in my opinion.

---

> ### Author Response · Authors · 2020-11-13
> **Anon Reviewer 3 Response**
>
> **We address each part reviewer's comments below:**
>
> *Quality
> -- The experiments are clear and the results are easily understandable.*
>
> *-- In all three experiments, the paper achieves compelling results with low errors compared to baselines, especially for high number of time steps.*
>
> **We thank the anonymous reviewer for the provided constructive criticism of the submitted work. The authors greatly appreciate the reviewer's effort to help us  improve the overall quality of the paper.**
>
> *Clarity
> --I find the mathematical notations in Section 2.2 could be more clear. I'm a bit confused about the infinite dimensionality of the state observables and how the Koopman operator handles it, whereas in the Figure 2 shown the Koopman operator K looks like a banded diagonal linear operator. It is not apparent why equation (3) leads to the construction of the encoder-decoder model shown. More explanations on the Koopman operator would be appreciated to make the paper more self-contained.*
>
> **In Section 2.2, we have elaborated on the core ideas of Koopman and emphasize that theoretically the Koopman observable is infinite-dimensional, but in practical applications a finite-dimensional approximation is used. We did not want to go too far in depth on the details of Koopman theory and analysis. We wanted our focus to be primarily on the  the transformer. We hope that the added details can make this section clearer for the reader.**
>
> *--The plots make the experiment results quite clear in terms of the model's capability to estimate the dynamics.*
>
> *--Some minor typos such as "on going" which should be "ongoing"*
>
> **Revised.  Thank you**
>
> *Originality
> -- This work uses a Transformer model instead of recurrent networks to perform dynamical system simulation. The paper also proposes using an embedding model. Overall, the novelty is moderate.*
>
> *Significance
> High-level pros and cons
> Pros*
>
> *-- Modeling dynamical systems is an interesting deep learning application.*
>
> *-- Good results compared to baselines in all experiments shown.*
>
>
> *Cons
> -- The clarity can be improved in my opinion.*
>
> **Thank you for your suggestions. We have made an effort to improve multiple parts of the paper to improve its clarity. This includes the addition of a schematic that should further clarify the training process.**

---

### Official Review · AnonReviewer4 · 2020-10-31
**A new application of transformer models (physical systems) with domain-specific embeddings**

**Rating:** 6
**Confidence:** 3

**Review:**

**Summary**:
The paper proposes applying transformer models to modeling physical systems. The state at each time step is embedded into a continuous vector using a pretrained encoder-decoder model based on Koopman’s theory. The experiments are performed on three physical systems and generally show that (1) a transformer model outperforms alternative machine learning methods, (2) a transformer model with the proposed embedding outperforms transformer models with alternative embeddings based on autoencoders or PCA, and (3) more transformer layers help (but only slightly).

**General comments**:
This is, to my knowledge, a new application area for transformer models, and the fact that they outperform alternatives could be interesting to the ML community.
Furthermore, the application of transformer models is not entirely standard, in that the paper proposes learning embeddings that are specialized to the application area.
The proposed approach is compared both with alternative (non-transformer) techniques and with ablations of the transformer model.

**Questions**:
The proposed embeddings are based on a neural network-based encoder and decoder pair and these are trained either with an objective based on Koopman’s theory or a standard AE objective. Have you considered training the embeddings jointly with the transformer parameters? I think this baseline is needed to motivate the need for pretrained embeddings and it could alleviate the concern that the model limitations are due to the "inaccuracies of the embedding model".

---

> ### Author Response · Authors · 2020-11-13
> **Anon Reviewer 4 Response**
>
> **We address each part reviewer's comments below:**
>
> *Summary: The paper proposes applying transformer models to modeling physical systems. The state at each time step is embedded into a continuous vector using a pretrained encoder-decoder model based on Koopman’s theory. The experiments are performed on three physical systems and generally show that (1) a transformer model outperforms alternative machine learning methods, (2) a transformer model with the proposed embedding outperforms transformer models with alternative embeddings based on autoencoders or PCA, and (3) more transformer layers help (but only slightly).*
>
> **The authors want to thank the reviewer for their thoughtful insights and recommendations to improve the quality of the paper. Their comments are greatly appreciated. The reviewer summarized the manuscript well.**
>
> *General comments: This is, to my knowledge, a new application area for transformer models, and the fact that they outperform alternatives could be interesting to the ML community. Furthermore, the application of transformer models is not entirely standard, in that the paper proposes learning embeddings that are specialized to the application area. The proposed approach is compared both with alternative (non-transformer) techniques and with ablations of the transformer model.
>
> Questions: The proposed embeddings are based on a neural network-based encoder and decoder pair and these are trained either with an objective based on Koopman’s theory or a standard AE objective. Have you considered training the embeddings jointly with the transformer parameters? I think this baseline is needed to motivate the need for pretrained embeddings and it could alleviate the concern that the model limitations are due to the "inaccuracies of the embedding model".*
>
> **The authors did indeed consider training the embedding model and transformer together as well as fine tuning the embedding model once the transformer has been trained. However we selected, that a pre-training approach of the embedding model would be a preferred approach for two main reasons as stated in the paper: the first is to remain as parallel as possible to NLP approaches (such as the use of word2vec and Byte Pair Encodings) to help support the idea that the default transformer is directly applicable to physical dynamics.  The second is to make the training of the transformer models more computationally tractable. Due to the potential domain sizes of physical systems and the size of modern transformers, training such transformers in the state space would prove to be very difficult with the needed memory requirements.**
>
> **The authors did however refactor the code base to test the reviewer's suggestion for the  Lorenz numerical test case. Using the same loss approach used to train the transformer but in the state-space (avoiding any back-prop through time), the authors found that jointly training the embedding model and transformer optimizes correctly but yields extremely poor predictions. We believe that this is largely due to the model simply disregarding the transformer/self-attention and predicting the reconstruction of only next time-step. Thus time-integration using the transformer in the embedding space does not work properly. We speculate that without a fixed embedding, it is likely a back-prop through time or sequential training methodology maybe required but this would be going against one of the  key benefits of the transformer.**
>
> **Regarding the comment of ``inaccuracies of the embedding model" in the paper, this was intended to refer to the required dimensionality reduction (from  65536 to a 512-dimensional vector). We believe this is fairly aggressive for the reaction diffusion system. We have added a note to make this  clear. Thank you.**

---

### Official Review · AnonReviewer2 · 2020-11-01
**a nice proposition, but the paper could be improved**

**Rating:** 7
**Confidence:** 5

**Review:**

This paper proposes the adaptation of the recent and SOTA framework
used in NLP to dynamical systems. The idea is nice and well
motivated. Two main steps are described: Transformer model for
markovian prediction of time series, along with a Koopman inspired
method to learn embeddings of the state space.

The interaction between the main training steps (embedding and
transformer parts) is not clear in the paper: pre-training of the
embeddings, followed by a fine tuning step (or not ?). This is an
issue of the paper since this interaction is important for NLP
applications. The reader should be able to fully understand what is
going on here, only by reading the paper.

My second concern is about the experimental part. The dynamical
systems are also limited in complexity, but this is not crucial here.
However, the difference in the results is not well documented and for
example, with the fluid flow behind a cylinder, the low results
obtained with CNN is surprinsing (too bad ?).

These concerns are not individually prohibitive for such a article,
and maybe the paper could be improved before
publication. Nevertheless, the ideas are really interesting.


In the introduction, authors claim that : "Standard deep neural
network architectures such as auto-regressive, recurrent and LSTM
based models have been largely demonstrated to be effective at
modeling various physical dynamics (...) However, the current literature is focused on systems of limited complexity and domain size."

It is maybe important here to better characterize the term complexity,
why it is limited ? and why it is important ?  Moreover, the
correlation between this complexity and the originality of using of
transformers is not straightforward. Complex architecture based on
LSTM for instance could be also effective or should better explain why
transformers are different when faced with larger complexity and
domain size.

The notations in the section 2 are a bit heavy and maybe it could be
easier to follow one of the experimental example, without loss of
generality, to introduce the whole picture.

---

> ### Author Response · Authors · 2020-11-13
> **Anon Reviewer 2 Response**
>
> **We address each part reviewer's comments below:**
>
> *This paper proposes the adaptation of the recent and SOTA framework used in NLP to dynamical systems. The idea is nice and well motivated. Two main steps are described: Transformer model for markovian prediction of time series, along with a Koopman inspired method to learn embeddings of the state space.*
>
> **The authors thank the reviewer for the well thought out review of the manuscript to improve the overall quality and clarity of the submitted work. The reviewer summarizes the paper well. When a revision is noted it will be reflected in the latest update of the paper.**
>
> *The interaction between the main training steps (embedding and transformer parts) is not clear in the paper: pre-training of the embeddings, followed by a fine tuning step (or not ?). This is an issue of the paper since this interaction is important for NLP applications. The reader should be able to fully understand what is going on here, only by reading the paper.*
>
> **Thank you for the suggestion. The authors agree that this should be more explicitly stated and have added a discussion prior to Section 2.1. Additionally, we have added a schematic that shows the two distinct training phases.**
>
> *My second concern is about the experimental part. The dynamical systems are also limited in complexity, but this is not crucial here. However, the difference in the results is not well documented and for example, with the fluid flow behind a cylinder, the low results obtained with CNN is surprinsing (too bad ?).
> These concerns are not individually prohibitive for such a article, and maybe the paper could be improved before publication. Nevertheless, the ideas are really interesting.*
>
> **The selected numerical examples are based on classical dynamical systems that are typically investigated in scientific machine learning. Namely, the Lorenz and the cylinder numerical examples. Thus we placed a priority on examples that  could directly resonate with the physical modeling and ML communities. The 3D Gray-Scott model was an additional choice to demonstrate the potential of this methodology to  more complex, higher-dimensional systems.**
>
> **Regarding the results of the CNN for the flow around a cylinder, we note that the convolutional model doesn't perform poorly in the sense that it is still able to produce physically accurate vortex shedding (one of the reasons we included Figs.13 and 14 in the appendix). The CNN model struggles with the initial transition from laminar to vortex shedding.  Due to the Markovian dependence, once the model deviates, it will continue to be inconsistent with the numerical solver.  We added a comment at the end of Section 4 to clarify this for the reader.**
>
> *In the introduction, authors claim that : "Standard deep neural network architectures such as auto-regressive, recurrent and LSTM based models have been largely demonstrated to be effective at modeling various physical dynamics (...) However, the current literature is focused on systems of limited complexity and domain size."
> It is maybe important here to better characterize the term complexity, why it is limited ? and why it is important ? Moreover, the correlation between this complexity and the originality of using of transformers is not straightforward. Complex architecture based on LSTM for instance could be also effective or should better explain why transformers are different when faced with larger complexity and domain size.*
>
> **We have elaborated on this in the introduction regarding the limitations of current ML models for predicting physical phenomena as well as  in the conclusion discussing when the transformer model has unique potential.**
>
> *The notations in  section 2 are a bit heavy and maybe it could be easier to follow one of the experimental examples, without loss of generality, to introduce the whole picture.*
>
> **The authors agree some of the notation is a bit dense in some areas, but we wanted to make sure everything is explicit. We have added additional details in the Koopman section to hopefully make these parts clearer for the reader. The added figure should also improves the clarity of this section. Thank you.**

---

### Decision · Program_Chairs · 2021-01-07
**Final Decision**

**Decision:**

Reject

**Comment:**

The paper introduces a framework for learning dynamical system models from observations consisting of discrete spatio-temporal series. It is composed of two components trained sequentially.  A first one learns embedding from observations using a seq2seq approach, where the embeddings are constrained to follow linear dynamics. This is inspired by approximation schemes for Koopman operators. These embeddings are then used as the spatio-temporal series representions and are fed to a transformer trained as an autoregressive predictor. Experiments are performed on different problems using data generated from PDEs through numerical schemes. Comparisons are performed with different ML baselines.

The paper is well written with experiments on problems with different complexities. The original contributions of the paper are 1) the combination of pretrained embeddings with a transformer auto-regressor, 2) a seq2seq architecture for learning time series representations constrained by linear dynamics.

On the cons side, the paper original contribution and significance are over claimed. Closely related ideas for learning approximate Koopman operators and observables have already been developed and used in similar contexts. Besides there is no discussion here on the properties or physical interpretability (which is often an argument for Koopman) of the learned representations. Then the baselines are mainly composed of simple regressors (LSTM, conv-LSTM, etc.) and this is not a surprise that they cannot learn dynamics over long term horizons. There is no comparison with dynamical models incorporating numerical integration schemes that could model the temporal dynamics of the system. There is now a large literature on this topic exploiting discrete (ResNet like) or continuous formulations (as started with Neural ODE).